# Bridging the Gap Between Offline and Online Reinforcement Learning Evaluation Methodologies

**Shivakanth Sujit**[12], **Pedro H. M. Braga**[123], **Jorg Bornschein**[4], **Samira Ebrahimi Kahou**[125]

[1]ÉTS Montréal, [2]Mila, Quebec AI Institute, [3]Universidade Federal de Pernambuco, [4]DeepMind,
[5]CIFAR AI Chair. Correspondence: shivakanth.sujit@gmail.com

## Abstract

Reinforcement learning (RL) has shown great promise with algorithms learning in environments with large state and action spaces purely from scalar reward signals. A crucial challenge for current deep RL algorithms is that they require a tremendous amount of environment interactions for learning. This can be infeasible in situations where such interactions are expensive; such as in robotics. Offline RL algorithms try to address this issue by bootstrapping the learning process from existing logged data without needing to interact with the environment from the very beginning. While online RL algorithms are typically evaluated as a function of the number of environment interactions, there exists no single established protocol for evaluating offline RL methods. In this paper, we propose a sequential approach to evaluate offline RL algorithms as a function of the training set size and thus by their data efficiency. Sequential evaluation provides valuable insights into the data efficiency of the learning process and the robustness of algorithms to distribution changes in the dataset while also harmonizing the visualization of the offline and online learning phases. Our approach is generally applicable and easy to implement. We compare several existing offline RL algorithms using this approach and present insights from a variety of tasks and offline datasets.

## 1 Introduction

Reinforcement learning (RL) has shown great progress in recent years with algorithms learning to play highly complex games with large state and action spaces such as DoTA2 ($\approx 10^4$ valid actions) and StarCraft purely from a reward signal of whether it won the game [Berner et al., 2019, Vinyals et al., 2019]. However, each of these breakthroughs required a tremendous amount of environment interactions, sometimes upwards of 40 years of accumulated experience in the game [Schrittwieser et al., 2019, Silver et al., 2016, 2018]. This can be infeasible for applications where such interactions are expensive, for example robotics. Offline RL methods tackle this problem by leveraging previously collected data to bootstrap the learning process towards a good policy. These methods can obtain behaviors that maximize rewards obtained from the system conditioned on a fixed dataset of experience. The existence of logged data from industrial applications provides ample data to train agents safely till they achieve good performance and then can be trained on real hardware. The downside of relying on offline data without any interactions with the system is that the behavior learned can be limited by the quality of data available [Levine et al., 2020].

Levine et al. [2020] point out that there is a lack of consensus in the offline RL community on evaluation protocols for these methods. The most widely used approach is to train for a fixed number of epochs on the offline dataset and report performance through the average return obtained over a number of episodes in the environment. In this paper, we propose to evaluate algorithms as a function of available data instead of just reporting final performance or plotting learning curves over a number of gradient steps. This approach allows us to study the sample efficiency and robustness of offline RL

algorithms to distribution shifts while also making it easy to compare with online RL algorithms as well as intuitively study online fine-tuning performance.

## 2 Background and Related Work

**Offline RL.** The simplest form of offline RL is behavior cloning which trains an agent to mimic the behavior present in the dataset, using the dataset actions as labels for supervised learning. However, offline datasets might have insufficient coverage of the states and operating conditions the agent will be exposed to. Such algorithms tend to be fragile and often perform poorly. In general, current offline RL algorithms address this issue by constraining the agent's policy around behavior seen in the dataset, for example, through enforcing divergence penalties on the policy distribution [Peng et al., 2019, Nair et al., 2020]. Another way of penalizing out of dataset predictions is by using a regularizer on the Q value to prevent actions that have low support in the data distribution from having high Q values as done in Conservative Q Learning (CQL) [Kumar et al., 2020]. This is a very brief overview of offline RL methods, and we direct readers to Levine et al. [2020] for a broader overview of the field.

**Metrics and Objectives.** The paradigm of *empirical risk minimization* (ERM) [Vapnik, 1991] is the prevailing training and evaluation protocol, both for supervised and unsupervised Deep Learning (DL). At its core, ERM assumes a fixed, stationary distribution and that we are given a set of (i.i.d.) data points for training and validation. Beyond ERM, and especially to accommodate non-stationary situations, different fields have converged to alternative evaluation metrics: In online learning, bandit research, and sequential decision making in general, the (cumulative) reward or the *regret* are of central interest. The regret is the cumulative loss accrued by an agent relative to an optimal agent when sequentially making decisions. For learning in situations where a single, potentially non-stationary sequence of observations is given, Minimum Description Length (MDL)[Rissanen, 1984] provides a theoretically sound approach to model evaluation. Multiple, subtly different formulations of the description length are in use, however, they are all closely related and asymptotically equivalent to *prequential MDL*, which is the cumulative log loss when sequentially predicting the next observation given all previous ones [Rissanen, 1987, Poland and Hutter, 2005]. Similar approaches have been studied and are called the *prequential approach* [Dawid and Vovk, 1999] or simply *forward validation*. A common theme behind these metrics is that they consider the agents' ability to perform well, not only in the big-data regime, but also its generalization performance at the beginning, when only a few observations are available for learning. The MDL literature provides arguments and proofs why those models, that perform well in the small-data regime without sacrificing their big-data performance, are expected to generalize better to future data [Rathmanner and Hutter, 2011].

With these two aspects in mind, that a) RL deals with inherently non-stationary data, and b) that sample efficiency is a theoretically and practically desirable property, we propose to evaluate offline RL approaches by their data efficiency.

## 3 Sequential Evaluation of Offline RL Algorithms

As mentioned above, one approach for offline RL evaluation is to perform multiple epochs of training over the dataset. We contend that there are a few issues with this approach. Firstly, this approach does not provide much information about the sample efficiency of the algorithm since it is trained on all data at every epoch. This means that practitioners do not see how the algorithm can scale with the dataset size, or if it can achieve good performance even with small amounts of logged data. Furthermore, there can be distribution changes in the quality of the policy in the dataset, and evaluating as a function of epochs hides how algorithms react to these changes. Finally, there is a disconnection in the evaluation strategies of online and offline RL algorithms, which can make it difficult to compare algorithms realistically.

Instead of treating the dataset as a fixed entity, we propose that the portion of the dataset available to the agent change over time and that the agents' performance is evaluated as a function of the available data. This can be implemented by reusing any of the prevalent replay-buffer-based training schemes from online deep RL. But instead of extending the replay-buffer with sampled trajectories from the currently learned policy, we instead slowly insert prerecorded offline RL data. We alternate between adding new samples to the buffer and performing gradient updates using mini batches

---

**Algorithm 1** Algorithm for Sequential evaluation in the offline setting.

---

1: **Input:** Algorithm $A$, Offline data $\mathcal{D} = \{s_t, a_t, r_t, s_{t+1}\}_{t=1}^{T}$, increment-size $\gamma$, gradient steps per increment $K$
2: Replay-buffer $\mathcal{B} \leftarrow \{s_t, a_t, r_t, s_{t+1}\}_{t=1}^{T_0}$
3: $t \leftarrow T_0$
4: **while** $t < T$ **do**
5:     Update replay-buffer $\mathcal{B} \leftarrow \mathcal{B} \cup \{s_t, a_t, r_t, s_{t+1}\}_{t}^{t+\gamma}$
6:     Sample a training batch, ensure new data is included: batch $\sim \mathcal{B}$
7:     Perform training step with $A$ on batch.
8:     $t \leftarrow t + \gamma$
9:     **for** $j = 1, \cdots, K$ **do**
10:        Sample a training batch $\sim \mathcal{B}$
11:        Perform training step with $A$ on batch.
12:    **end for**
13: **end while**

---

sampled from the buffer. The number of samples added to the buffer at a time is denoted by $\gamma$ and the number of gradient steps performed between each addition to the buffer is denoted by $K$. A concrete implementation of the approach is outlined in Alg. 1.

This approach of evaluation addresses several of the issues with epoch-style training. By varying $\gamma$ and $K$ we can get information about the scaling performance of an algorithm with respect to dataset size, which tells us if data is the bottleneck for further improvements, and how quickly the algorithm can learn with limited data. We can visualize how the algorithm behaves with shifts in dataset quality directly from the performance curves. There is a direct analogy for evaluation in the online RL setting since online methods are evaluated as a function of the number of environment steps, which is a measure of the amount of data it has access to in the replay buffer and hence can be directly connected to the size of the replay buffer in the offline method. We can also seamlessly evaluate the performance of the algorithm in online fine-tuning by adding samples from the environment once the entire offline dataset is added to the replay buffer. A benefit of the sequential approach is that it does not require a complete overhaul in codebases that follow existing training paradigms. For the baselines that we present in this paper, we were able to use the sequential evaluation approach with less than 10 lines of changes to the original codebases.

**Implementation Details**   To ensure that the algorithm sees each data point in the dataset at least once, when a new batch of data is added to the buffer, the algorithm is trained on that sample of data once before $K$ mini-batches are sampled from the buffer for training. If the batch added is smaller than the mini-batch size, then it can be made part of the next mini-batch that is trained on. In practice, we found that setting $\gamma$ and $K$ to 1 worked well in all datasets tested. This means that the x-axis of all plots directly corresponds to the number of samples available for training and the number of gradient updates performed. The changes made to the codebase of each algorithm are as follows: Each codebase had a notion of a replay buffer that was being sampled, and the only addition required here was a counter that kept track of up to which index in the buffer data points could be sampled from to create mini-batches. The counter was initialized to $T_0 = 5000$ so that there were some samples in the buffer at the start of training. The second change that needed to be made was changing the outer loop of training from epochs to the number of gradient updates and incrementing the buffer counter by $\gamma$ every $K$ update. This way, the amount of data the algorithm was trained on sequentially increased to the full dataset over the course of training.

## 4   Experiments

We evaluate several existing offline RL algorithms using the sequential approach, namely IQL [Kostrikov et al., 2022], CQL [Kumar et al., 2020], TD3+BC [Fujimoto and Gu, 2021], AWAC [Nair et al., 2020] and Behavior Cloning (BC). These algorithms were evaluated on the D4RL benchmark [Fu et al., 2020], which consists of three environments: Halfcheetah-v2, Walker2d-v2 and Hopper-v2. For each environment, we evaluate four versions of the offline dataset: random, medium, medium-expert, and medium-replay. Random consists of 1M data points collected using a random

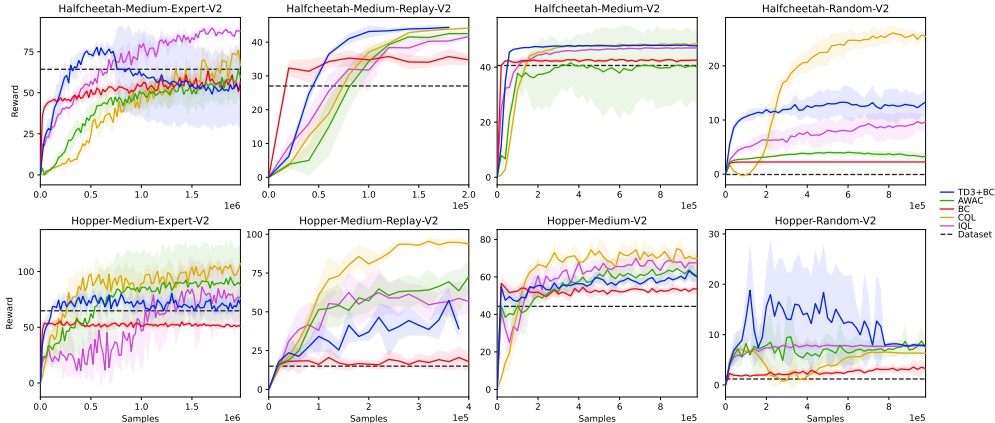

Figure 1: Performance curves on the D4RL benchmark of offline RL algorithms as a function of data points seen. Shaded regions represent standard deviation across 5 seeds.

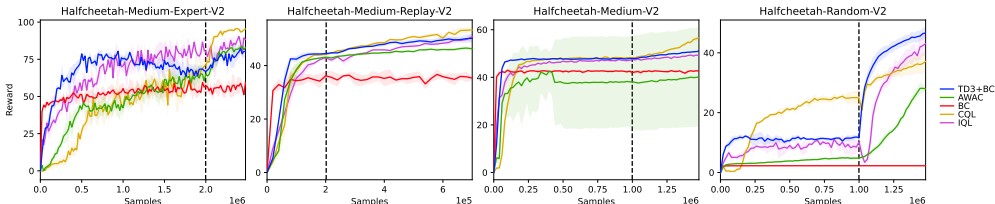

Figure 2: Performance curves for online fine-tuning. Each algorithm is given 500k steps in the simulator after sequential evaluation of the offline dataset. Dotted line indicates where online fine-tuning begins. Shaded regions represent standard deviation across 3 seeds.

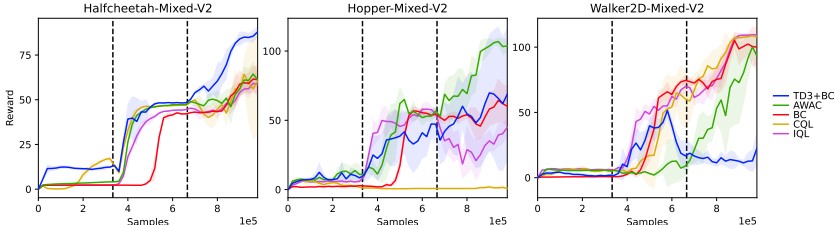

Figure 3: Performance curves for Mixed dataset with varying dataset quality. Dotted line indicates where there is a change in the dataset generating policy distribution. Horizontal dotted line indicates the performance of the policy that generated the data. Shaded regions represent standard deviation across 5 seeds.

policy. Medium contains 1M data points from a policy that was trained for one-third of the time needed for an expert policy, while medium-replay is the replay buffer that was used to train the policy. Medium-expert consists of a mix of 1M samples from the medium policy and 1M samples from the expert policy. These versions of the dataset are useful for evaluating the performance of offline agents across a wide spectrum of dataset quality. The experiments on a subset of datasets are given in Fig. 1 and the complete set of datasets, along with an experiment comparing curves with larger $K$ are available in the appendix. In each plot, we also include the performance of the policy that generated the dataset as a baseline, which provides context for how much information each algorithm was able to extract from the dataset. This baseline is given as a horizontal dotted line.

We also show how sequential evaluation supports seamless integration of online fine-tuning experiments into performance curves. In this setting, once the entire offline dataset is added to the replay buffer, the agent is allowed to interact with the online simulator for a fixed number of steps (500k steps in our experiments). Since the curves are a function of data samples, we can continue evaluating

performance as before. The results on a subset of datasets are given in Fig. 2, and curves for all datasets are available in the appendix. Finally, to highlight how sequential evaluation can visualize how the algorithm reacts to changing dataset quality during training, we create a "mixed" version of each environment in which the first 33% of data comes from the random dataset, the next 33% from the medium dataset and the final 33% from the expert dataset. Each algorithm is sequentially given samples and from the performance curves given in Fig. 3 we can see how each algorithm adapts to changes in the dataset distribution.

For each dataset, we train algorithms following Alg. 1, initializing the replay buffer with 5000 data points at the start of training. We set $\gamma$ and $K$ each to 1, that is, there is one gradient update performed each time a sample is added to the buffer. The results are presented in Fig. 1, where the x-axis represents the amount of data in the replay buffer.

One striking observation from Fig. 1 is how quickly the algorithms converge to a given performance level and then stagnate. This is most evident in the medium version of each environment. With less then 300K data points in the buffer, each algorithm stagnates and does not improve in performance beyond that point even after another 500K points are added. There are diminishing returns from adding data beyond 500K points to the buffer. This highlights that most of the tested algorithms are not very data-hungry. That is, they do not require a large data store to reach good performance, which is beneficial when they need to be employed in practical applications. The experiment highlights that the chosen datasets might lack diversity in collected experience since most algorithms appear to need only a fraction of it to attain good performance. In 5 of the 8 datasets in Fig. 1, CQL reaches better performance than other methods earlier in training and consistently stays at that level as more data is added. TD3+BC exhibits an initial steep rise in performance but levels out at a lower score overall, or in the case of Halfcheetah-medium-expert, degrades in performance as training progresses. Even in online fine-tuning, CQL reaches higher scores compared to the other algorithms showing the versatility of CQL in handling both offline datasets and online interactions with the system. In the mixed dataset experiment, there is no clear lead in performance across environments. While CQL adapts quickly in both HalfCheetah and Walker2d, it fails to learn at all in Hopper. TD3+BC outperforms all other methods in HalfCheetah, but is the worse performing in Walker2d. A surprising result is how well BC performs in each environment, with BC nearly being the second best performing algorithm in all environments.

## 5 Conclusion

In this paper, we propose a sequential style of evaluation for offline RL methods so that algorithms are evaluated as a function of data rather than compute or gradient steps. In this style of evaluation, data is added sequentially to a replay buffer over time, and mini-batches are sampled from this buffer for training. This is analogous to online training to deep RL and allows us to measure the data scaling and robustness of offline algorithms simultaneously from the training curves. We compared several existing offline methods using sequential evaluation and showed how their training curves allow for algorithm selection depending on data efficiency or performance. We believe that sequential evaluation holds promise as an established method of evaluation for the offline RL community. Future work in this domain could explore the effect of $\gamma$ and $K$ on algorithms and their ramifications. One drawback of sequential evaluation is that it assumes that there exists a simulator that can be cheaply used to evaluate the agent periodically. This may not always be possible, and in those cases, off-policy evaluation methods are used [Thomas et al., 2015, Wang et al., 2020].

## Acknowledgments and Disclosure of Funding

The authors would like to thank the Digital Research Alliance of Canada for compute resources, NSERC, Google and CIFAR for research funding.

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

# A  Description of datasets

The D4RL locomotion benchmark consists of three environments with varying data quality. The size of each version and the performance of the policy that generated it is given in Table 1.

Table 1: Dataset sizes in D4RL

| Dataset | Size | Average score in Dataset |
|---|---|---|
| halfcheetah-random-v2 | 999000 | -0.07 |
| halfcheetah-medium-v2 | 999000 | 40.60 |
| halfcheetah-medium-replay-v2 | 201798 | 27.03 |
| halfcheetah-medium-expert-v2 | 1998000 | 64.29 |
| halfcheetah-expert-v2 | 999000 | 87.91 |
| walker2d-random-v2 | 999999 | 0.01 |
| walker2d-medium-v2 | 999322 | 61.94 |
| walker2d-medium-replay-v2 | 301698 | 14.81 |
| walker2d-medium-expert-v2 | 1998318 | 82.55 |
| walker2d-expert-v2 | 999000 | 106.90 |
| hopper-random-v2 | 999999 | 1.19 |
| hopper-medium-v2 | 999998 | 44.28 |
| hopper-medium-replay-v2 | 401598 | 14.97 |
| hopper-medium-expert-v2 | 1998966 | 64.78 |
| hopper-expert-v2 | 999061 | 108.24 |

# B  Performance on D4RL Benchmark

We present complete training curves on all twelve datasets that were used in Fig. 4 and final performance in Table 2. In addition to the curves, we compare the algorithms at the end of training with scores aggregated across environments. This is done using the rliable [Agarwal et al., 2021] library to plot interval estimates of normalized performance measures such as median, mean, interquartile mean (IQM) and optimality gap. The optimality gap is a measure of how far an algorithm is from optimal performance aggregated across environments. So, lower values are better. The scores in each dataset are normalized with respect to the maximum score, which is 100. These results are given in Fig. 6. In both the performance curves and aggregated scores we can see that CQL outperforms other tested methods by a clear margin. A curious phenomenon observed was that AWAC [Nair et al., 2020] is unable to learn at all in the Walker2d environment with either the medium-expert or the medium version achieving very low rewards. As a sanity check we set $\gamma$ to the size of the dataset and reran experiments and the method achieved results similar to those reported in the paper, indicating it was not an implementation problem. This is surprising since AWAC is proposed as an algorithm that can work for online fine-tuning following offline pretraining, but among all the methods tested, it had drastic changes in performance when using sequential evaluation. The final performance in the online fine-tuning task and the mixed version of the environment is also given in Tables 3 and 4.

Moreover, Fig. 7 show how the algorithms performed when $K$ was set to 2. This experiment studied if we had not trained the methods for enough gradient steps and if additional performance could be extracted from the data. However, performance remained essentially the same or even degraded in some instances, showing that this was not the case.

Table 2: Performance of each algorithm on the D4RL Benchmark

| Dataset | TD3+BC | AWAC | BC | CQL | IQL |
|---|---|---|---|---|---|
| halfcheetah-medium-expert-v2 | 6.83 | 59.04 | 63.44 | 83.45 | 86 |
| halfcheetah-medium-replay-v2 | 44.26 | 43.37 | 36.3 | 43.54 | 41.97 |
| halfcheetah-medium-v2 | 48.45 | 46.58 | 43.12 | 49.13 | 46.96 |
| halfcheetah-random-v2 | 10.09 | 2.26 | 2.26 | 24.43 | 8.41 |
| hopper-medium-expert-v2 | 69.92 | 111.88 | 44.53 | 111 | 49.18 |
| hopper-medium-replay-v2 | 63.04 | 65.47 | 14.02 | 88.51 | 69.09 |
| hopper-medium-v2 | 49.77 | 52.8 | 55.84 | 70.53 | 67.49 |
| hopper-random-v2 | 8.13 | 9.18 | 2.26 | 6.2 | 7.63 |
| walker2d-medium-expert-v2 | 108.33 | 1.98 | 107.58 | 109.98 | 98.51 |
| walker2d-medium-replay-v2 | 76.8 | 82.54 | 24.63 | 73.31 | 63.12 |
| walker2d-medium-v2 | 83.4 | 1.76 | 78.78 | 83.16 | 80.85 |
| walker2d-random-v2 | 0.49 | 3.48 | 0.63 | -0.12 | 7.82 |

Table 3: Performance of each algorithm in the online fine-tuning task on the D4RL Benchmark

| Dataset | TD3+BC | AWAC | BC | CQL | IQL |
|---|---|---|---|---|---|
| finetune-halfcheetah-medium-expert-v2 | 74.36 | 83.6 | 61.77 | 96.72 | 87.74 |
| finetune-halfcheetah-medium-replay-v2 | 50.52 | 46.87 | 28.66 | 53.25 | 49.53 |
| finetune-halfcheetah-medium-v2 | 51.9 | 52.89 | 42.86 | 55.29 | 49.24 |
| finetune-halfcheetah-random-v2 | 48.53 | 30.67 | 2.26 | 34.17 | 50.87 |
| finetune-hopper-medium-expert-v2 | 112.71 | 111.9 | 48.01 | 100.25 | 110.66 |
| finetune-hopper-medium-replay-v2 | 92.56 | 65.56 | 13.19 | 102.78 | 100.38 |
| finetune-hopper-medium-v2 | 63.03 | 45.45 | 58.92 | 94.31 | 73.59 |
| finetune-hopper-random-v2 | 9.85 | 8.78 | 2.47 | 4.56 | 12.05 |
| finetune-walker2d-medium-expert-v2 | 107.77 | 1.41 | 108.35 | 110.73 | 110.81 |
| finetune-walker2d-medium-replay-v2 | 77.78 | 94.11 | 5.61 | 89.77 | 95.71 |
| finetune-walker2d-medium-v2 | 85.01 | 21.32 | 66.47 | 83.7 | 84.24 |
| finetune-walker2d-random-v2 | 8.6 | 1.67 | 0.5 | 0.33 | 10.56 |

Table 4: Performance of each algorithm in the mixed version of the D4RL Benchmark

| Dataset | TD3+BC | AWAC | BC | CQL | IQL |
|---|---|---|---|---|---|
| halfcheetah-mixed-v2 | 80.65 | 29.55 | 57.38 | 93.47 | 68.04 |
| hopper-mixed-v2 | 112.42 | 110.36 | 86.67 | 0.75 | 51.37 |
| walker2d-mixed-v2 | 8.15 | 98.36 | 108.79 | 109.71 | 109.83 |

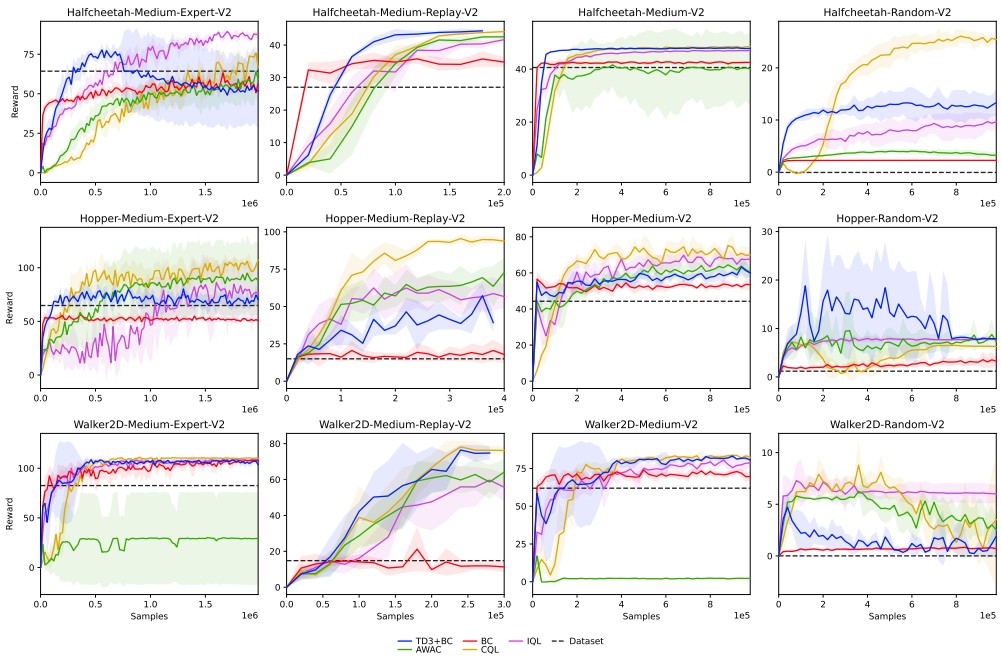

Figure 4: Performance curves on the D4RL benchmark of offline RL algorithms as a function of data points seen. Shaded regions represent standard deviation across 5 seeds.

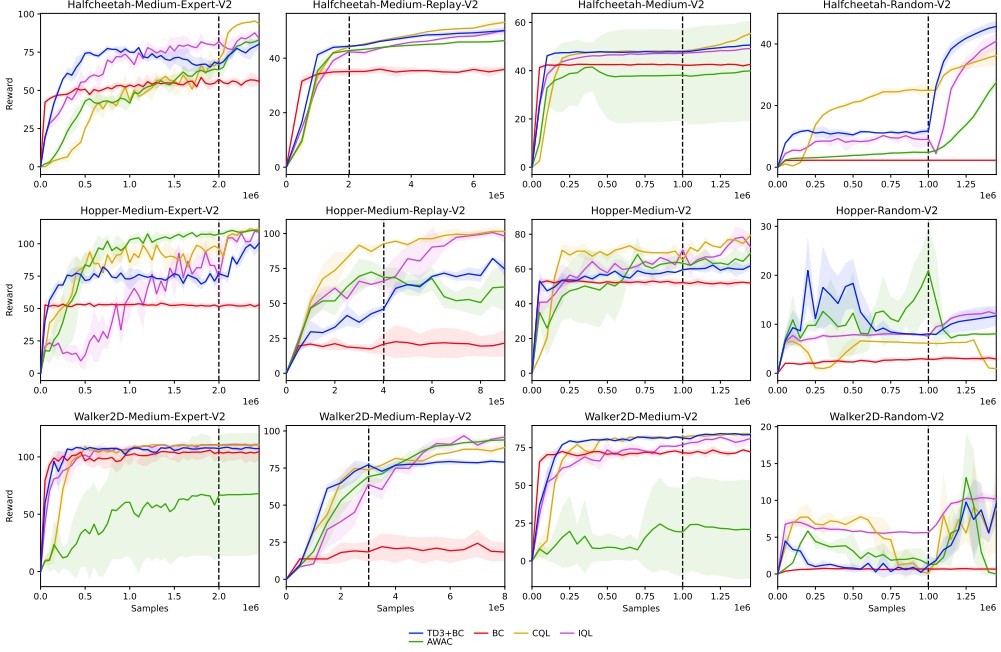

Figure 5: Performance curves for online fine-tuning. Each algorithm is given 500k steps in the simulator after sequential evaluation of the offline dataset. Dotted line indicates where online fine-tuning begins. Shaded regions represent standard deviation across 3 seeds.

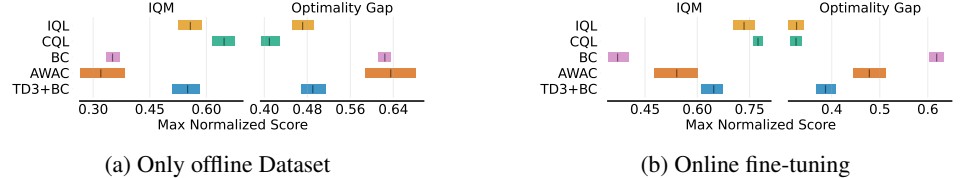

(a) Only offline Dataset          (b) Online fine-tuning

Figure 6: Performance aggregated across environments using rliable. For IQM higher is better, while for Optimality gap, lower is better

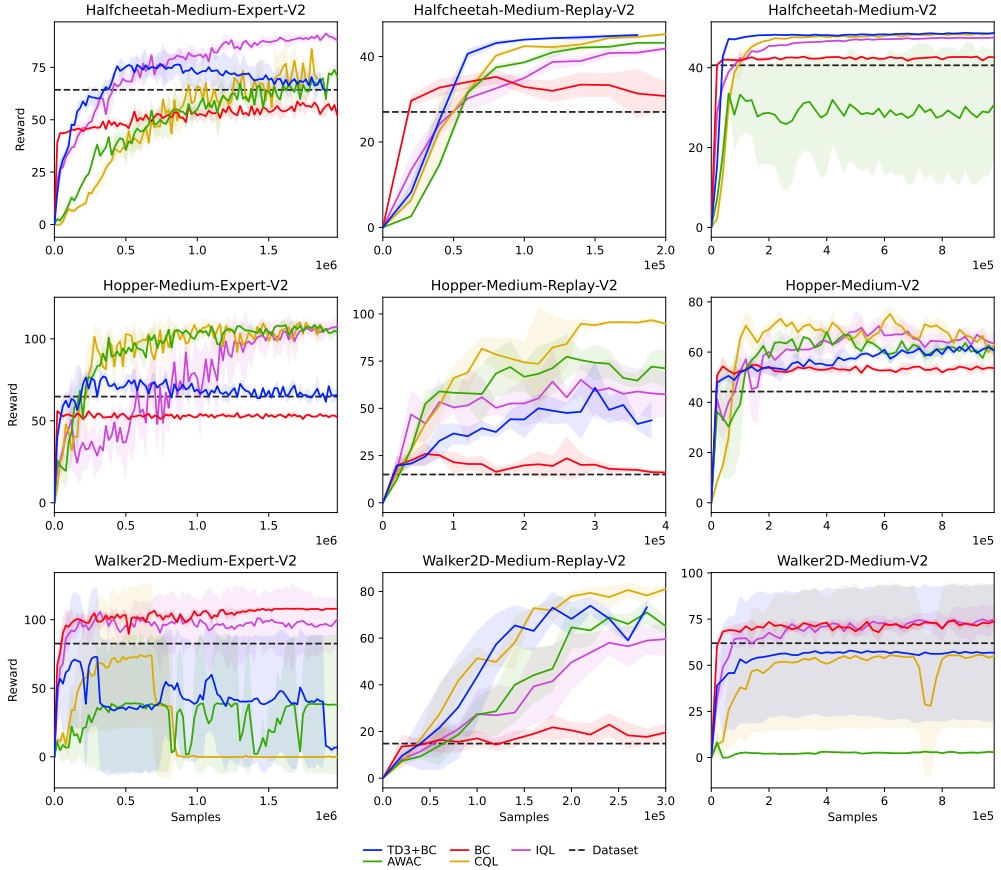

Figure 7: Performance curves on the D4RL benchmark with $K$ increased to 2. Shaded regions represent standard deviation across 3 seeds.

