# OpenReview forum: "Bridging the Gap Between Offline and Online Reinforcement Learning Evaluation Methodologies"
_NeurIPS.cc/2022/Workshop/Offline_RL — Offline RL Workshop NeurIPS 2022_

### Official Review · Reviewer_BbbY · 2022-10-17
**Review of "Bridging the Gap Between Offline and Online Reinforcement Learning Evaluation Methodologies"**

**Rating:** 6
**Confidence:** 3

**Review:**

This paper presents an evaluation methodology for offline reinforcement learning algorithms. Contrary to the most common paradigm where policies are trained on a dataset for many epochs and then finally evaluated, this paper proposes to instead evaluate methods based on their performance as the size of the offline dataset increases. Specifically, a set number of trajectories from the offline dataset are added into the replay buffer at specific intervals of gradient steps.

Strengths:
- I think the idea of a sequential evaluation of offline RL algorithms is quite interesting, and could be helpful to understand other aspects of offline RL performance including sample efficiency and performance on data of varying quality.
- The strategy is simple to implement, as the authors mention.
- The writing in the paper is clear and the experiments are thorough.

Weaknesses:
- As the paper itself mentions, one challenge of a sequential evaluation procedure such as this is that it requires access to an evaluation environment throughout training, or off-policy evaluation. This is challenging if the real-world use case of offline RL is one in which simulated or real evaluations are expensive or dangerous to collect. However, I think this could still be useful as an analysis tool for offline RL algorithms so that algorithm and data selection can happen with the aid of simulators and then applied to other situations.
- One other concern is that the sequential evaluation introduces additional hyperparameters (gamma and K), which are the rate at which additional data points are added to the dataset and the number of gradient steps taken between each addition of data points. Because different algorithms may require different numbers of gradient steps to train on the same size of data points, this may lead to an additional value that needs to be tuned for the sequential evaluation. Even if one is interested in just sample efficiency of a particular algorithm, they still must tune the value of K to ensure that they are achieving the best possible performance. This can lead to a large computational cost.
- A remark on the title is that I feel that it is a bit misleading to indicate that there is a “gap” between offline and online reinforcement learning evaluation methodologies. Given the different problem statements, it is reasonable for the evaluation methodologies to also be different, and I don’t think it’s obvious that existing offline RL evaluation methodologies are worse than online ones?

Despite the concerns raised above, I believe that alternative evaluation protocols for offline RL are an interesting topic and to my knowledge relatively unexplored in the community.

---

### Official Review · Reviewer_tGyv · 2022-10-18
**A new Offline RL evaluation approach, but lagging in a few ways**

**Rating:** 6
**Confidence:** 5

**Review:**

Summary:

The paper proposes a new offline RL evaluation approach, which sequentially adds the existing offline data to the agent's replay buffer and periodically evaluates agent performance as a function of the available data.

Discussion:

I believe the proposed approach lacks in a few ways:

1. The main issue with current evaluation schemes is that they require access to the real environment, which is the anti-thesis of offline RL, particular in high-stakes environments (i.e. autonomous driving, financial, medical applications). The current method does not address this issue and moreover would require significantly higher number of environment evaluations.

2. The proposed approach is justified by the non-stationarity of the RL problem, however offline RL is largely stationary, the given dataset does not suffer from distribution shift, which only appears at evaluation time. The proposed method could be more relevant to the fine-tuning regime.

3. The proposed methodology could not reflect algorithm strengths and performance accurately. I believe there are two modes to consider here:
- The dataset is time-indexed, i.e. similar to an online-RL replay buffer or there is a shift in the data-generating policy (Figure 3) and added sequentially. This is largely not a realistic scenario for the current state of approaches (i.e. learning from human data, sub-optimal demonstrations, play data, etc..). Moreover many offline algorithms explicitly or implicitly estimate data supports, hence would perform poorly in this non-stationary environment, they were not designed for it (versus algorithms designed for fine-tuning performance).
- The dataset is more uniform, in which case we are just adding additional data from within the same data distribution. It is not immediately clear what benefits this evaluation approach would have. It does not reflect "data efficiency" in the classical sense, since this is a question of data density, rather that data quality/coverage.

Overall I believe the proposed method is of limited practical use in more realistic data regimes or applications, but nevertheless this is an important topic to discuss in the community and the proposed method could be a valuable starting point.